# Mining Candidate Genes Related to Heavy Metals in Mature Melon (*Cucumis melo* L.) Peel and Pulp Using WGCNA

**DOI:** 10.3390/genes13101767

**Published:** 2022-09-30

**Authors:** Qi Shen, Xiaonan Wu, Yongxia Tao, Guorong Yan, Xian Wang, Shuangyu Cao, Cheng Wang, Weizhong He

**Affiliations:** 1Xinjiang Key Laboratory of Agricultural Product Quality and Safety/Agricultural Product Quality and Safety Risk Assessment Laboratory of the Ministry of Rural Agriculture/Institute of Agricultural Quality Standards and Testing Technology, Xinjiang Academy of Agricultural Sciences, Urumqi 830091, China; 2School of Food Science and Pharmacy, Xinjiang Agricultural University, Urumqi 830091, China; 3New Plant Variety Testing (Urumqi) Branch Center of the Ministry of Agriculture and Rural Affairs, Crop Variety Resources Institute, Xinjiang Academy of Agricultural Sciences, Urumqi 830091, China; 4Xinjiang Key Laboratory of Agricultural Product Quality and Safety/Agricultural Product Quality and Safety Risk Assessment Laboratory of the Ministry of Rural Agriculture, Xinjiang Academy of Agricultural Sciences, Urumqi 830091, China

**Keywords:** WGCNA, weighted gene co-expression network analysis, melon, heavy metal ions, RNA-seq, candidate gene

## Abstract

The content of metal ions in fruits is inseparable from plant intake of trace elements and health effects in the human body. To understand metal ion content in the fruit and pericarp of melon (*Cucumis melo* L.) and the candidate genes responsible for controlling this process, we analyzed the metal ion content in distinct parts of melon fruit and pericarp and performed RNA-seq. The results showed that the content of metal ions in melon fruit tissue was significantly higher than that in the pericarp. Based on transcriptome expression profiling, we found that the fruit and pericarp contained elevated levels of DEGs. GO functional annotations included cell surface receptor signaling, signal transduction, organic substance metabolism, carbohydrate derivative binding, and hormone-mediated signaling pathways. KEGG pathways included pectate lyase, pentose and glucuronate interconversions, H^+^-transporting ATPase, oxidative phosphorylation, plant hormone signal transduction, and MAPK signaling pathways. We also analyzed the expression patterns of genes and transcription factors involved in hormone biosynthesis and signal transduction. Using weighted gene co-expression network analysis (WGCNA), a co-expression network was constructed to identify a specific module that was significantly correlated with the content of metal ions in melon, after which the gene expression in the module was measured. Connectivity and qRT–PCR identified five candidate melon genes, *LOC103501427*, *LOC103501539*, *LOC103503694*, *LOC103504124,* and *LOC107990281*, associated with metal ion content. This study provides a theoretical basis for further understanding the molecular mechanism of heavy metal ion content in melon fruit and peel and provides new genetic resources for the study of heavy metal ion content in plant tissues.

## 1. Introduction

Melon is an annual vine herb of Cucurbitaceae cucumber, belonging to diploid plants also known as cantaloupe, an economically important crop [1]. Melon flesh is thick, rich in aroma, sweet, and rich in nutrients. It contains various sugars, organic acids, vitamins, and minerals, and is consumed as a fruit. At the same time, it has additional medicinal effects, such as analgesic, anti-inflammatory, and antioxidant effects, as well as anticancer, diuretic, anti-hypothyroidism, atherosclerosis prevention, and other beneficial effects [2,3,4,5,6].

Heavy metals are a collective term that refers to metals and metalloids with a density greater than 4 ± 1 g/cm³ [7]. Compared with other pollutants, the heavy metal pollution of soils is particularly difficult to ameliorate because metals entering the soil cannot be degraded and are difficult to remove naturally; therefore, heavy metals accumulate in the soil and can eventually harm human health through the food chain and even destroy the soil-crop ecosystem [8,9,10]. For example, Cd entering the food chain is an endocrine disruptor that can cause damage to organs such as the kidney, liver, and lungs, and has carcinogenic and teratogenic effects [11,12]. Arsenic (As) can enter the human body through breathing, drinking water, and food. When the limit is exceeded, it will cause acute and chronic poisoning in the human body, resulting in skin cancer, lung cancer, bladder cancer, and other diseases [13,14]. Heavy metals can denature and inactivate important enzymes or proteins, interfere with electron transfer during photosynthesis and respiration, stimulate the production of reactive oxygen species, cause peroxidation of cells, and activate the initiation of plant defense mechanisms [15,16]. Although some elements are beneficial to the human body in trace amounts, if the amount that enters the body through food and drinking water exceeds a safe dose, it can cause potential harm to the body. Some metal ions can be bioaccumulated in the human body and even be converted into more harmful organometallic compounds [17].

With the rapid development and wide application of next-generation sequencing (NGS) technology, RNA-seq analysis has become an effective means to study gene expression and metabolic, gene regulatory networks, providing a basis for the identification of candidate genes involved in specific processes [18]. RNA-seq technology has been widely used in the study of various biotic or abiotic stresses. Based on microarray or RNA-seq expression data, weighted gene co-expression network analysis (WGCNA) constructs a scale-free topological overlap matrix to describe relationships between genes and then divides genes with similar expression patterns into groups in a gene expression module [19]. WGCNA is used to study the biological correlation between co-expressed gene modules relate to target traits and to explore core genes in gene co-expression networks. As a typical systems biology method, WGCNA has been widely used in botanical research. Tan et al. analyzed the transcriptome data of 17 rice strains (*Oryza sativa*) at different time points of cadmium treatment and identified 22 gene modules. Combined with differential expression analysis, a total of 164 genes related to the cadmium stress response were mined [20]. Zou et al. performed WGCNA on the fiber transcriptome data of two cotton lines at different developmental stages, identified five specific modules related to fiber development, and mined the core genes in the modules [21]. From the transcriptome data of 33 light-treated eggplant peel samples, SmWRKY44 was found to be significantly associated with the accumulation of anthocyanins in eggplant peels, leaves, and flowers based on WGCNA and module correlation [22].

Various molecular mechanisms allow plants to cope with heavy metal stress and protect their normal growth and metabolism [23]. While growing, melons are at risk of being contaminated by heavy metals, and the taste and quality of melon contaminated by heavy metals have undergone critical changes. At present, research on melons mainly focuses on the quality, color, and taste of the fruit, and there are few studies on heavy metal ions [24,25]. To study the content of heavy metal ions and their candidate genes in distinct parts of melon fruit and pericarp tissue, we first measured the As, Cd, Cr, Ni, and Pb content in various parts of melon fruit and pericarp tissue and then performed RNA-seq sequencing. The data were subjected to differential expression analysis, GO and KEGG enrichment analysis, and expression analysis of hormone-related genes and TFs. By constructing a weighted gene co-expression network, candidate genes related to heavy metal ion content in melon fruit and pericarp tissue were identified. This study provides a theoretical basis for further understanding the molecular mechanisms of heavy metal ion uptake in melon fruit and peel and provide new genetic resources for the study of heavy metal ion content in melon and other plants.

## 2. Materials and Methods

### 2.1. Plant Materials

In July 2020, the melon material Xizhoumi No. 25 was planted in Lukeqin Town, Shanshan County, Turpan City, Xinjiang, China. In September 2020, six parts of the peel (A, B, C) and pulp (D, E, F) were collected from three locations of the melon fruit (Figure 1a). Tissue samples collected at separate locations were quickly frozen in liquid nitrogen, transported to the laboratory, and stored in a −80 °C freezer. Three biological replicates were prepared for each sample.

### 2.2. Heavy Metal Content Detection

Inductively coupled plasma–mass spectrometry (ICP–MS) was used to detect the content of As, Cd, Cr, Ni, and Pb in six parts of melon fruit peel and pulp at distinct positions [26]. Fresh samples of melon fruit, peel, and pulp were weighed (1 g), placed in a microwave digestion inner tank with 5 to 10 mL of nitric acid, loosely covered and incubated overnight, after which the lid was tightened and then digested according to the standard operating procedures of the microwave digestion apparatus. After cooling, the tank cover was slowly opened to exhaust, and the inner cover was rinsed with a small amount of water. The digestion tank was placed on a temperature-controlled electric hot plate or an ultrasonic water bath, heated at 100 °C for 30 min, brought up to 50 mL with water and mixed well. The blank solution and the sample solution were injected into the inductively coupled plasma mass spectrometer. The signal response values of the element to be measured and the internal standard element were measured, and the concentration of the heavy metal element to be measured in the digestion solution was obtained according to the standard curve.

### 2.3. RNA Sequencing

RNA-seq library construction was performed on 18 samples from 3 biological replicates in 6 sites in 2 tissues of mature melon, and the quantity of RNA was confirmed using an Agilent 2100 Bioanalyzer (Agilent, Santa Clara, CA, USA) before library construction. and quality assessment, the standard was that the RNA integrity number must be >7, and the rRNA ratio (28S/18S) must be >1.5 [27]. Total mRNA (0.1–0.4 µg) was copurified and fragmented using PCR plates with magnetic plate holders. Fragment mRNA was reverse transcribed to cDNA using Superscript II and random primers (Invitrogen, Carlsbad, CA, USA), and RNA sequencing (RNAseq) was performed on the IlluminaHiSeq 2000 platform (BGI, China) [28].

### 2.4. RNA-Seq Data Analysis

Fastp software was used to perform quality control on the raw data obtained by sequencing, such as removing adapters, removing primer sequences, filtering reads with a high N content of unknown bases, and obtaining high-quality clean reads [29]. The clean reads were aligned to melon’s reference genome (https://www.ncbi.nlm.nih.gov/assembly/GCF_000313045.1, accessed on 1 May 2021.) using HISAT2, and String Tie quantified the aligned reads [30]. FPKM (fragments per kilobase of exon per million fragments mapped) refers to the number of reads mapped to each kilobase of exons in the reads per million mapped, using FPKM (fragments per kilobase of exon per million fragments mapped) to measure gene expression [31].

### 2.5. Identification, Annotation and Cluster Analysis of DEGs

The DEGs (differentially expressed genes) were screened, and DESeq2 was used to calculate the differential expression fold of genes between different samples based on gene expression. The absolute value of FDR ≤ 0.01 and log2-fold change ≥ 1 were used to filter DEGs [32]. To further understand the possible functions of DEGs and to explore the biological functions related to the accumulation of heavy metals in various positions of melon fruit peel and pulp, the DEGs obtained from each comparison group were analyzed by Gene Ontology (GO) and Kyoto Encyclopedia of Genes and Genomes (KEGG) enrichment analysis and visualized using the R language ggplot2 package [33]. K-means cluster analysis was performed using the R language pheatmap package, and line graphs of expression patterns were drawn using the ggplot2 package [34].

### 2.6. Hormone Biosynthesis, Signal Transduction and Transcription Factor Analysis

The amino acid sequences of all DEGs were submitted to the KEGG (https://www.kegg.jp/ghostkoala/, accessed on 1 May 2021.) database for the identification of hormone biosynthesis and signal transduction genes, and their expression levels were visualized using the R language pheatmap. The protein sequences of DEGs were submitted to PlantTFDB (http://planttfdb.gao-lab.org/tf.php?sp=Ppe&did=Prupe.I004500.1.p, accessed on 1 May 2021.) for analysis and prediction, thereby obtaining differentially expressed transcription factors [35].

### 2.7. WGCNA

The gene expression profile of DEGs was analyzed by the dynamic branch-cut method using the R language WGCNA package. To ensure the distribution of the scale-free network, the weighting coefficient β should satisfy the correlation coefficient close to 0.9 and have a certain degree of gene connectivity. In this study, β = 5 was chosen as the weighting factor [36]. In the automatic network construction function, block wise modules are used to construct the network, multiple valid modules are obtained, and the number of genes contained in each module is not equal. With minModuleSize = 30 and Merge Cut Height = 0.25 as the standard, the modules with a similarity of 0.75 were merged. The correlation coefficient between the characteristic vector ME (Module Eigengene) of the module and the content of trace elements was calculated. Specificity modules were screened with r > 0.80 and *p* < 0.05 as criteria. The genes and predicted transcription factors in the specific modules were extracted to construct the co-expression network, and Cytoscape_v3.8 software was used to visualize the co-expression network [37].

### 2.8. qRT–PCR

According to the cDNA information of the gene, Primer 5.0 software was used to design primers in the specific region of the gene sequence (Appendix A). Total RNA from 6 parts of two tissues of melon was extracted by an RNA extraction kit (Beijing, China, Baimaike), and the extracted RNA was reverse transcribed using a reverse transcription kit (Baimaike). qRT–PCR was performed using a Roche LightCycler96, and the amplification volume was 20 μL. The reaction program was pre-denaturation at 94 °C for 30 s, denaturation at 95 °C for 5 s, annealing at 60 °C for 5 s, and extension at 72 °C for 35 s, for 39 cycles. The results were analyzed for relative quantification using the 2^–ΔΔCt^ method. Each sample was repeated three times, and the internal reference gene was CmActin.

## 3. Results

### 3.1. Determination of Metal Ion Content in Melon Peel and Fruit Tissue

The content of heavy metals in soil has increased sharply, and the problem of plants being polluted by heavy metals is serious. To study the content of heavy metals in various parts of melon after ripening, we selected six distinct parts of the peel and pulp for sampling and measured the content of heavy metal ions by inductively coupled plasma–mass spectrometry (ICP–MS) (Figure 1a). Through these measurements, we found that the content of As, Cd, Cr, Ni, and Pb in the peel was significantly higher than that in the pulp tissue, and in the peel, the content of metal ions in Part B was the lowest, whereas the content of metal ions in Parts D, E and F in the pulp was the lowest. There was little difference (Figure 1b). This indicates that the difference in metal ion content between tissues of melon is greater than that of various parts of the same tissue.

### 3.2. Global Transcriptome Changes in Melon Peel and Fruit Tissue

To further investigate the mechanism underlying the differences in metal ion content in various parts of melon fruit and pericarp, we performed RNA-seq sequencing on three biological replicate samples from six parts. After filtering the original data, a total of 127.09 Gb of clean data was obtained. The average clean data of each sample reached 6.87 Gb, the percentage of Q30 bases was over 86.74%, the GC content was greater than 44.32%, and the alignment rate was greater than 93.18%. The above data show that the quality of the transcriptome data was qualified (Appendix A).

First, correlation analysis between samples was performed (Figure 2a), and it was found that the correlations between the three replicates in each treatment period were all over 0.90, indicating that the correlation between replicates was high. However, the three parts of the pericarp and the three parts of the fruit were obviously separated, indicating that the difference between the pericarp and the fruit was greater than the difference between various parts of the same tissue, which was consistent with the results of the analysis of metal ion content. PCA was then performed (Figure 2b), which was consistent with the correlation analysis results. In summary, the correlation between repetitions is high, and the periods are clustered together. To confirm the accuracy of transcriptome expression profiles, 10 genes were randomly selected for qRT–PCR analysis of three independent replicates in six tissues, and the results and RNA-Seq expression patterns showed similar trends, indicating that the RNA-Seq data were dependable and suitable for further analysis (Appendix A).

### 3.3. Difference Analysis

Since the content of metal ions in Part B is more stable than that in Parts A and C, we used the B part of pericarp tissue as a control to perform differential expression analysis using DESeq2 and calculated the gene expression by FPKM value. Compared with site B, the number of DEGs at site A was 656, the number of DEGs at site C was 643, the number of DEGs at site D was 2158, the number of DEGs at site E was 1919, and the number of DEGs at site F was 2019 (Figure 3a). There were 96 DEGs between B and A, 191 DEGs between B and C, 233 DEGs between B and D, 261 DEGs between B and E, and 261 DEGs between B and F. There were 157 DEGs. The number of DEGs shared across the 5 sites was 117 (Figure 3a,b). The number of DEGs at sites A, C, and B was less than that at sites D, E, F, and B, which further proves that the difference between tissues is greater than that in various parts of the same tissue, which is consistent with the previous correlation and PCA results.

### 3.4. GO and KEGG Enrichment Analysis of DEGs

The results of DEG enrichment analysis showed (Figure 4) that the biological processes annotated by GO functions mainly included cell surface receptor signaling, signal transduction, organic substance metabolic process, carbohydrate derivative binding, hormone-mediated signaling pathway, and auxin-activated signaling pathway (Figure 4a). The cellular component mainly included intracellular membrane-bound organelles, response to auxin, polymeric cytoskeletal fibers, phragmoplasts, supramolecular polymers, and supramolecular fibers. The molecular functions mainly included hydrolase activity, microtubule motor activity, oxidoreductase activity, oxidizing metal ions, carbohydrate transmembrane transporter activity, ATP-dependent microtubule motor activity, glucosyltransferase activity, and exopeptidase activity. KEGG pathways were mainly pectate lyase, pentose and glucuronate interconversions, H^+^-transporting ATPase, oxidative phosphorylation, sucrose-phosphate synthase, starch and sucrose metabolism, V-type H^+^-transporting ATPase subunit, plant hormone signal transduction, MAPK signaling pathways, and fatty acid degradation (Figure 4b).

### 3.5. Expression Pattern Analysis of DEGs

Genes with the same biological function may have the same expression pattern, and a total of five statistically significant clusters were identified using k-means clustering. To gain further insight into their functional transitions, we used the k-means clustering algorithm to group all 3146 DEGs, including 243 transcription factors, and then performed expression pattern analysis (Figure 5a). The k-means cluster analysis highlighted an overall burst of gene transcriptional up- or downregulation in melon fruit and pericarp. Most of the clusters exhibited distinct pulsatile transient changes at the transcriptional level (Figure 5b). The expression of Cluster 1 in fruit was significantly higher than that in pericarp, and the expression in Part B was the lowest. The expression of Cluster 2 in pericarp is significantly higher than that in fruit, and the expression in Part B is the highest. The overall expression pattern of Cluster 3 and Cluster 1 was basically the same, but in the pericarp, the expression in the three parts was basically the same, and the expression in the E part was the highest. The highest expression of Cluster 4 was mainly in A and C, and the expression in B, D, E, and F was basically the same. Cluster 5 is the highest in B and E, and the expression in A, C, D, and F is generally lower.

### 3.6. Expression Analysis of Hormone-Related Genes

Hormones play a vital role in plant growth and development and stress. We analyzed the expression profiles of all hormone biosynthesis- and signal transduction-related DEGs in distinct parts of melon fruit and pericarp. We first analyzed the expression profiles of ABA biosynthesis and signal transduction genes. ABA biosynthesis genes were mainly highly expressed in the B site, and signal transduction genes were relatively highly expressed in fruit tissues (Figure 6a). MeJA biosynthesis and signal transduction genes were mainly expressed at relatively prominent levels in the pericarp (Figure 6b). Genes for auxin biosynthesis and signal transduction were mainly expressed at relatively prominent levels in the pericarp (Figure 6c). Genes for gibberellin biosynthesis were relatively highly expressed mainly in the fruit, and genes for signal transduction were relatively highly expressed in the B site (Figure 6d). The genes for ethylene biosynthesis and signal transduction were mainly expressed at relatively elevated levels in the B site of the peel (Figure 6e). The salicylic acid biosynthesis genes were mainly highly expressed in the B site, and the signal transduction genes were relatively highly expressed in the pericarp. The gene expression of cytokinin biosynthesis and signal transduction is complex, which may also be due to the close relationship between cytokinin and fruit ripening (Figure 6g).

### 3.7. Expression Analysis of Specific TFs

Transcription factors (TFs) are major regulators of gene expression. To further investigate the expression patterns of transcription factors, we performed transcription factor predictions on all DEGs and obtained a total of 243 differentially expressed transcription factors, mainly MYB, NAC, WRKY, AP2, bHLH, and C2H2 (Figure 7a). Most had the highest expression at site B, and only a few were highly expressed in both the pericarp and the fruit (Figure 7b–g). This heatmap shows the expression patterns of differentially expressed TF genes (Figure 7b–g). Among them, most AP2 had the highest expression in Part B, and a small portion had the highest expression in fruit (Figure 7b). Most of the MYB expression levels were highest in the pericarp, and a small portion of the expression was highest in the fruit (Figure 7c). Half of C2H2 had the highest expression at site B, and half of C2H2 had the highest expression at site B (Figure 7d). The expression of WRKY was highest mainly in the B site (Figure 7e). Most of the NACs had the highest expression in the pericarp, and a small portion of them had the highest expression in the fruit (Figure 7f). The expression of bHLH was highest in most of the pericarp, and a small part was highest in the fruit (Figure 7g).

### 3.8. Mining of Candidate Genes Related to WGCNA and Metal Ion Content

To reveal the function of the network, rather than the function of individual genes, we used a systems biology approach weighted gene co-expression network analysis (WGCNA) on 3146 differentially expressed genes to construct a co-expression network of various parts of melon that contained a similar A module of gene composition for expression patterns. The β soft threshold is set to 5, the unscaled R2 > 0.80, the dynamic shearing algorithm is used to calculate the correlation coefficient between DEGs, and the matrix is constructed for clustering and module division. Different modules have assorted colors, and finally, a total of six expression modules are obtained. (Figure 8a, Appendix A). To further understand the relationship between gene expression and metal ion content and to identify candidate genes related to metal ion content, according to the correlation results between expression modules and six parts, the brown module was significantly and highly correlated with metal ion content (Figure 8b). Brown was selected for constructing gene interaction networks and hub screening. To identify the major hub genes of these modules, the gene network was visualized using Cytoscape (Figure 8c). This module contains three types of transcription factors, and a total of six hub genes were identified. To further explain the relationship between the six hub genes and the metal ion content, we used qRT–PCR to examine their expression patterns at the six sites (Appendix A). Among them, *LOC103501427*, *LOC103501539*, *LOC103503694*, *LOC103504124* and *LOC107990281* were higher in pericarp tissue than in fruit tissue, with the highest expression in Part B of pericarp, whereas *LOC103502364* had almost the same expression in fruit and pericarp (Appendix A). We speculate that *LOC103501427*, *LOC103501539*, *LOC103503694*, *LOC103504124* and *LOC107990281* may be related to the content of metal ions in melon fruit and peel and can be used as candidate genes to further study their functions in regulating the content of melon metal ions and to analyze their molecular mechanisms.

## 4. Discussion

Due to the influence of various human activities, heavy metal and metalloid pollution has increasingly become a worldwide environmental problem [38]. Studies by Heavy et al. have shown that when plants are stimulated by heavy metals, they not only restrict the entry of heavy metals into plant cells and roots but also eventually cause the heavy metal content in plants to exceed the standard [39]. In most plants, Cd, as a nonessential element, can accumulate competitively against the uptake of certain essential metals, hindering vegetative growth and reducing crop quality [40]. At present, to increase the output of melon and meet market demand, the cultivation area of melon is increasing, and the abuse of organic and inorganic fertilizers is common, resulting in the accumulation of soil nitrate nitrogen and vegetable nitrate content and heavy metal elements, which poses a great threat to the agricultural environment and a potential threat to food safety [41]. The content of heavy metals in soil has increased sharply, and the problem of heavy metal contamination of plants is serious [39]. By measuring the metal content of the peel and distinct parts of the fruit, we found that the content of As, Cd, Cr, Ni, and Pb in the peel was significantly higher than that in the pulp tissue (Figure 1). Plant organic cation transporters help the excretion of endogenous cations for the absorption and elimination of cation toxicity. The reason the content of metal ions in melon pulp is low may be that the heavy metals in the parts with high heavy metal concentrations seriously inhibit the conformational change of the ion transporter during the transport process, thereby inhibiting the activity of the transporter and making the ion transport effect. It may also be that the peel of melon has a strong absorption effect on heavy metals.

In recent years, transcriptome sequencing has become an efficient, fast, and low-cost high-throughput sequencing technology that is widely used in numerous studies [18,19,20,21,42]. RNA-Seq technology was used to identify and characterize the expression of a large number of genes, and algorithms were combined to develop genome-wide coexpression networks to provide new genes for molecular breeding. A study found candidate genes related to maize plant height by RNA-seq analysis of three different maize hybrid combinations [43]. RNA-seq analysis of *Arabidopsis thaliana* under moderate drought conditions revealed important roles of hormone-related genes in plant metabolism and signal transduction during dehydration responses [44].

When plants encounter heavy metal stimuli, they trigger signal transduction, including hormone signal transduction. During hormone transduction, hormones respond to gene expression regulation by targeting specific transcription factors and gene promoter regions [45]. Some studies have found that Cd can respond to heavy metal stress in plants through plant hormones, such as ethylene, salicylic acid, auxin, and jasmonic acid [46,47,48,49]. Ethylene can modulate oxidative stress pathways that increase with increasing H_2_O_2_ content, and jasmonic acid and salicylic acid can play a protective role by alleviating Cd-induced oxidation [46,47,48,49]. At low concentrations of heavy metals, auxin can reduce toxicity in plants by reducing the production of H_2_O_2_ and inhibiting the adsorption of Cd [50]. When plants are under heavy metal stress, such as that caused by Cd, they increase flavonoid production, which can function in antioxidant systems with Peroxidase, Ascorbate peroxidase (APX), etc., to alleviate the peroxidation of membranes, thereby maintaining the normal metabolism of the organism [46]. Through the expression analysis of hormone biosynthesis and signal transduction and TF-related genes, we found that most of the genes were overexpressed in the B site in the peel, which indicated that they may also play the same role in melon fruit (Figure 6 and Figure 7). Our enrichment analysis showed that cell surface receptor signaling, organic substance metabolic processes, carbohydrate derivative binding, hormone-mediated signaling pathway, auxin-activated signaling, hydrolase activity, microtubule motor activity, oxidoreductase activity, metal ion oxidation, pectate lyase, pentose and glucuronate interconversions, H^+^-transporting ATPase, oxidative phosphorylation, sucrose-phosphate synthase, and starch and sucrose metabolism were also significantly associated with the content of metal ions.

In recent years, many genes involved in the heavy metal response have been identified and speculated in plants [51,52,53]. The expression of these genes plays an important role in the transport and enrichment of heavy metal ions in cells and the improvement of plant resistance. The study found that the expression of AtMYB50 and AtMYB61 in Arabidopsis thaliana was strongly inhibited by lead regardless of short-term or long-term induction, and the loss-of-function mutants of AtMYB50 and AtMYB61 genes had significantly lower lead content than the wild type [54]. The expression of the tomato NAC transcription factor SlNAC063 was upregulated by Al stress. In addition, the expression level of SlNAC063 induced by Al increased with increasing Al concentration and time [52]. SlNAC063 negatively regulates tomato aluminum tolerance by CRISPR/Cas9 gene editing technology [52]. The Arabidopsis thaliana WRKY33 gene can be induced to express by exogenous Cd stress, and the Cd accumulation in the overexpression line is lower than that of the wild type, indicating that WRKY33 can regulate the tolerance of Arabidopsis thaliana to Cd stress [55]. The interaction between bHLH121 and bHLH IVc transcription factor coactivates the expression of FIT to regulate the iron response in Arabidopsis thaliana [56]. We also found that transcription factors such as MYB, NAC, WRKY, AP2, bHLH, and C2H2 (Figure 7a) may be related to the content of heavy metal ions in melon, providing reliable candidate genes for further research on the content of neutron number ions in melon. Of course, if these TFs can be further functionally verified by transgenic and CRISPR/Cas9 gene editing technology in the future, they will provide reliable functional genes for molecular breeding.

WGCNA is widely used in the clarification of coexpression network modules. By calculating the connectivity of genes in a module, the position and importance of the gene in the network can be inferred [42]. We selected the top six genes with the highest connectivity in the specificity module as core genes and speculated that they may be related to the heavy metal content of melon fruit and peel. The expression of *LOC103501427*, *LOC103501539*, *LOC103503694*, *LOC103504124*, and *LOC107990281* in pericarp tissue was higher than that in fruit tissue as shown using qRT–PCR, and the highest expression was in the B part of pericarp, whereas *LOC103502364* was almost the same in fruit and pericarp. The function in melon is not yet clear and further research must be conducted using biotechnological methods such as overexpression identification and gene knockout. Genes with a higher degree of connection to core genes may also be significantly related to the heavy gold content of melon fruit and peel, which also provides potential candidate genes for later research and lays the foundation for further analysis of its molecular mechanisms.

## 5. Conclusions

In conclusion, we performed expression analysis and WGCNA of hormone biosynthesis and signal transduction-related genes based on RNA-Seq data by measuring the content of heavy metal ions in different parts of melon peel and fruit. The results of enrichment analysis showed that the differentially expressed genes mainly regulated the tolerance of melon to heavy metal ions through cell surface receptor signaling, signal transduction, pectate lyase, pentose and glucuronate interconversions, and H^+^-transporting ATPase pathways. MYB, NAC, WRKY, AP2, bHLH, and C2H2 transcription factors were also found to be important candidate genes related to heavy metal ion content in melons. By constructing a weighted gene co-expression network, a specificity module that was significantly correlated with the metal ion content of melon tissues was identified, and connectivity was used as an indicator to reveal the specific genes that may play key roles in the specificity module. The results of this study identified candidate genes for further study of metal ions in melons and also provide theoretical guidance for research on similar molecular and biochemical mechanisms in other plants.

## Figures and Tables

**Figure 1 genes-13-01767-f001:**
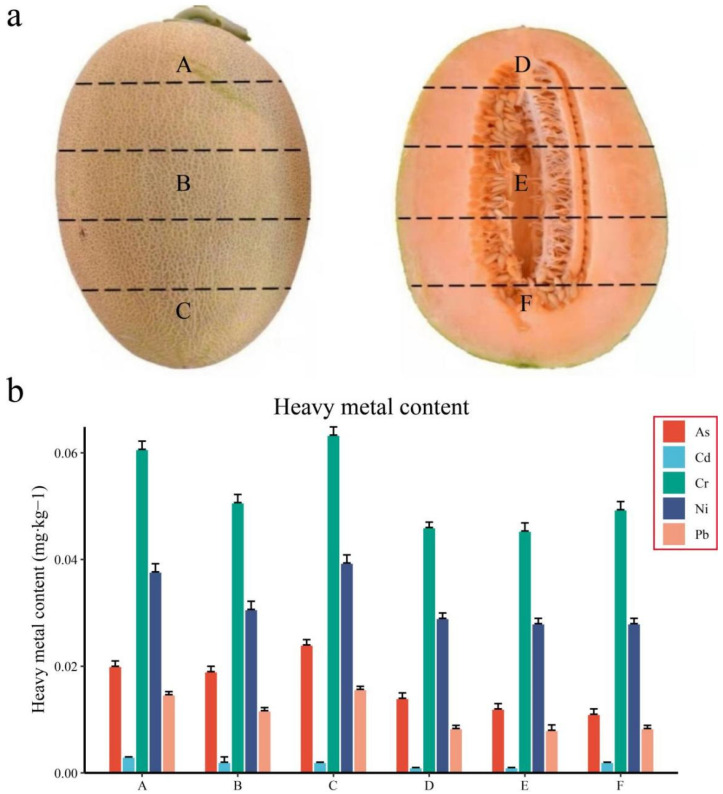
Schematic diagram of sampling organization and content of various heavy metals. (**a**) Six sampling sites of melon peel and pulp, (**b**) As, Cd, Cr, Ni, and Pb content of six sampling sites.

**Figure 2 genes-13-01767-f002:**
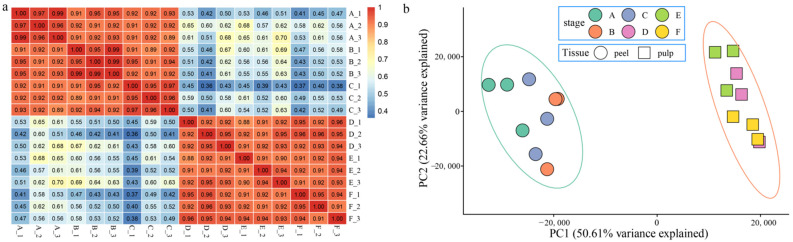
Correlation of samples and PCA. (**a**) Correlation analysis of samples from various parts of melon fruit and peel. (**b**) Principal component analysis (PCA) for correlation analysis of samples from various parts of melon fruit and peel.

**Figure 3 genes-13-01767-f003:**
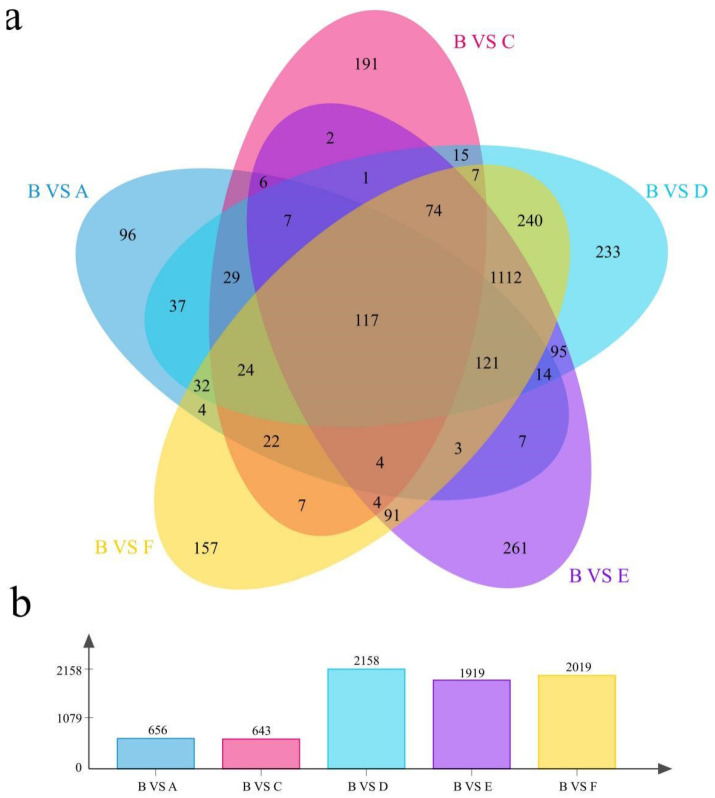
Differential analysis of transcriptomes in various parts of melon fruit and pericarp. (**a**) Number of common and unique DEGs in various parts. (**b**) Number of DEGs.

**Figure 4 genes-13-01767-f004:**
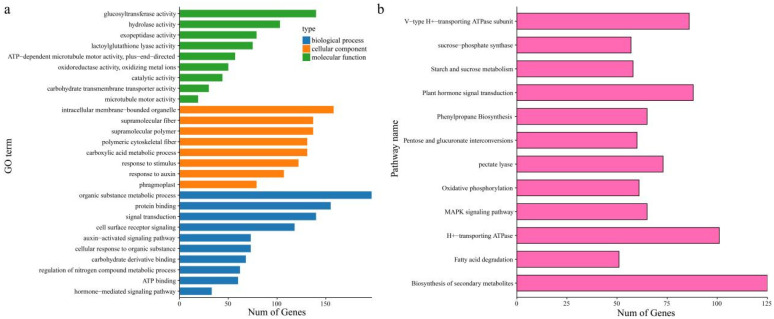
Differentially expressed gene enrichment analysis. (**a**) GO enrichment analysis results of differentially expressed genes. (**b**) KEGG pathway enrichment analysis results of differentially expressed genes.

**Figure 5 genes-13-01767-f005:**
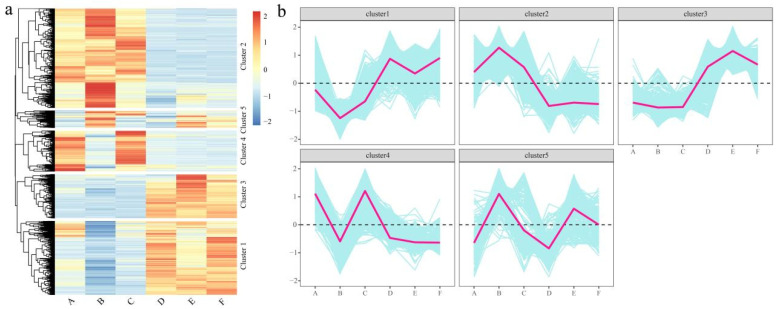
Analysis of differentially expressed gene expression patterns. (**a**) Hierarchical clustering results of differentially expressed genes. (**b**) Line graph of gene expression trends for each cluster.

**Figure 6 genes-13-01767-f006:**
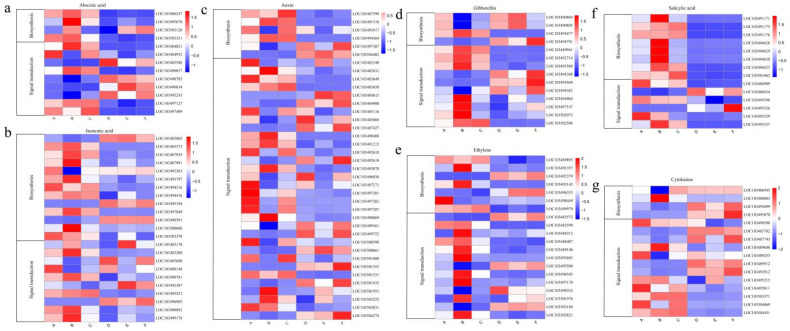
Expression analysis of hormone-related genes. (**a**) Differential expression of genes for ABA biosynthesis and signal transduction at different sample; (**b**) differential expression of genes for MeJA biosynthesis and signal transduction at different sample; (**c**) differential expression of genes for Auxin biosynthesis and signal transduction at different sample; (**d**) differential expression of genes for Gibberellin biosynthesis and signal transduction at different sample; (**e**) differential expression of genes for Ethylene biosynthesis and signal transduction at different sample; (**f**) differential expression of genes for Salicylic acid biosynthesis and signal transduction at different sample; (**g**) differential expression of genes for Cytokinine biosynthesis and signal transduction at different sample.

**Figure 7 genes-13-01767-f007:**
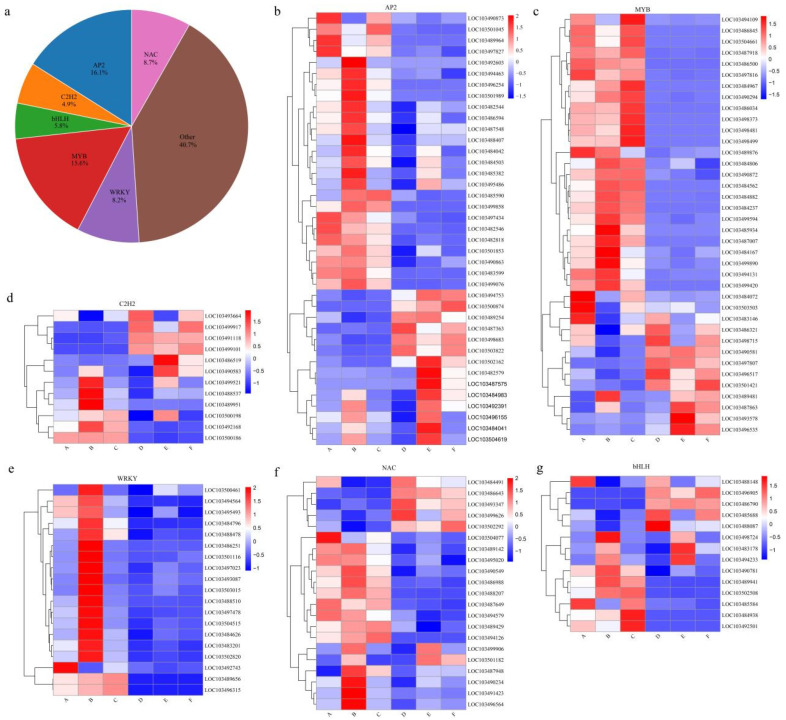
Expression analysis of specific transcription factors. (**a**) The ratio of core differentially expressed transcription factors; (**b**) the expression profile of AP2 transcription factor; (**c**) the expression profile of MYB transcription factor; (**d**) the expression profile of C2H2 transcription factor; (**e**) the expression profile of WRKY transcription factor expression profile; (**f**) expression profile of NAC transcription factor; (**g**) expression profile of bHLH transcription factor.

**Figure 8 genes-13-01767-f008:**
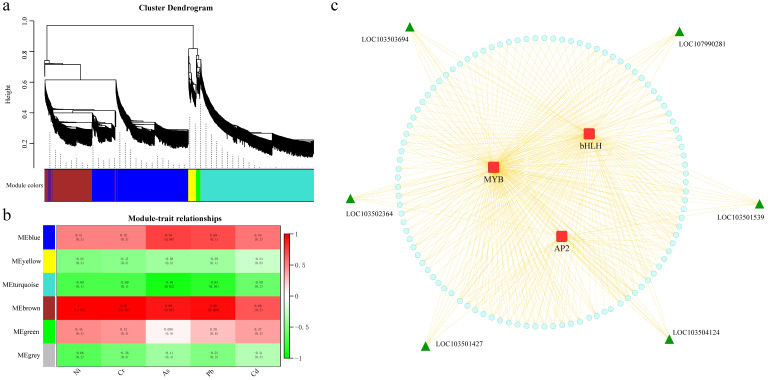
(**a**) Hierarchical clustering tree (dendrogram) of genes based on co-expression network analysis; (**b**) heatmap of correlation and significance between modules and metal ion content; (**c**) correlation specificity of heavy metal ion content within the module The gene co-expression network.

## Data Availability

The data of the article has been uploaded to the NCBI SRA database (https://www.ncbi.nlm.nih.gov/sra, accessed on 27 September 2022); the Accession number is PRJNA859241.

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
