# Peer review of "Mining Candidate Genes Related to Heavy Metals in Mature Melon (Cucumis melo L.) Peel and Pulp Using WGCNA"

_genes, 2022, doi:10.3390/genes13101767_

Round 1

Reviewer 1 Report

The study on mining gene for heavy metal in melon is quite interesting and is of present day need.

1. Author had focused on DEG based on RNA sequencing of sample collected randomly from different site. It is just a theoretical approach However, this study would have been scientifically planned in a better way using contrasting genotypes along with various crop stage treatment of heavy metal or they could used different varieties rather using only a single variety at different locations/sites.

2. How author is sure about a variety with low metal content not contained all studied DEG?

3. RT-PCR validation picture should be included as supplementary figure

4. It just a theoretical assumption in my view author plan with various stages of crop to cover all the metal related gene/TFs 

5. In reference section few reference as mentioned in text are missing and author must use a uniform style of references. 

6. Authors should address all highlighted point before consideration.

Author Response

Dear reviewer,

Thank you very much for your comments on our manuscript. According to your comment, we have revised our manuscript and identified it in red and have updated it in this revision. The responses we provided are marked in red.

The study of mining genes for heavy metals in melon is quite interesting and is of present day need.

Point 1: The author focused on DEGs based on RNA sequencing of samples collected randomly from different sites. It is just a theoretical approach However, this study would have been scientifically planned in a better way using contrasting genotypes along with various crop stage treatment of heavy metal or they could used different varieties rather using only a single variety at different locations/sites.

Response 1: Thanks for this comment. We selected materials of different genotypes for research and found that Xizhoumi No. 25 had the most obvious difference in the content of heavy metal ions in different tissues in the mature stage. There are also insignificant changes. We have also constructed a genetic population based on these materials and are also conducting corresponding research. Melons are mainly eaten after they are ripe, so there was no research on different growth periods at that time, and later, we will design experiments to conduct research on different development periods.

Point 2: How author is sure about a variety with low metal content not contained all studied DEG?

Response 2: Thanks for this insightful comment. Thank you very much for your suggestion. We used the site with the lowest heavy metal ion content as a control to conduct differential analysis to obtain all DEGs. We also calculated the correlation between all differentially expressed genes and heavy metal ion content based on WGCNA to obtain the corresponding candidate genes.

Point 3: RT‒PCR validation picture should be included as supplementary figure

Response 3: Thanks for this comment. This section has been revised and updated in this revision.

Point 4: It just a theoretical assumption in my view author plan with various stages of crop to cover all the metal related gene/TFs.

Response 4: Thanks for your constructive comments. We led to the analysis of all differential TFs and found some comparative core TFs (MYB, NAC, WRKY, AP2, bHLH and C2H2) that may be related to the heavy metal ion content of melon. Of course, the response of these transcription factors to heavy metal ions has been well studied in Arabidopsis and rice, which also shows that our results are reliable. We also analyzed the expression pattern of TF to provide important candidate genes for subsequent related studies.

Point 5: In reference section few reference as mentioned in text are missing and author must use a uniform style of references.

Response 5: Thanks for this comment. Thank you very much for your suggestions. This section has been revised and updated in this revision.

Point 6: Authors should address all highlighted point before consideration.

Response 6: Thanks for this comment. Thank you very much for your suggestions. This section has been revised and updated in this revision.

Reviewer 2 Report

I believe that the manuscript presents valuable results which are applicable to many other plant species.

The results are well presented.

However, the discussion section needs some amendments and additions from a more in-depth molecular point of view to address the results of the present experiment. 

The conclusion section is not enough informative. Rewrite the section.

Author Response

Dear reviewer,

Thank you very much for your comments on our manuscript. According to your comment, we have revised our manuscript and identified it in red and have updated it in this revision. The responses we provided are marked in red.

Point 1: I believe that the manuscript presents valuable results that are applicable to many other plant species.The results are well presented. However, the discussion section needs some amendments and additions from a more in-depth molecular point of view to address the results of the present experiment.

Response 1: Thanks for your valuable comment. We have revised the discussion section. This section has been revised and updated in this revision.

Point 2: The conclusion section is not enough informative. Rewrite the section.

Response 2: Thanks for your constructive comments. We have revised the conclusions section. This section has been revised and updated in this revision.

Round 2

Reviewer 1 Report

Author should use FDR≤0.01 and log2-fold change≥2  and then identify the unique DEGs, as log2-fold change ≥1 may not give you significant DEGs for the study.

if author can add MDS plot for the study 

author is asked to add current the ref:  https://doi.org/10.1007/s13205-020-02416-w in the text file 

Author Response

(The authors gave the same response as above.)
